# Improvement of RT-DETR model for ground glass pulmonary nodule detection

**Siyuan Tang**[1,2☯], **Qiangqiang Bao**[3☯], **Qingyu Ji**[4], **Tong Wang**[2], **Naiyu Wang**[2], **Min Yang**[2], **Yu Gu**[3], **Jinliang Zhao**[3], **Yuhan Qu**[2], **Siriguleng Wang**[1]*

1 College of computer science and technology, Inner Mongolia Normal University, Hohhot, Inner Mongolia, China, 2 Baotou Medical College, Inner Mongolia University of Science and Technology, Baotou, Inner Mongolia, China, 3 School of Digital and Intelligence Industry, Inner Mongolia University of Science and Technology, Baotou, Inner Mongolia, China, 4 Second Affiliated Hospital of Baotou Medical College, Inner Mongolia University of Science and Technology, Baotou, Inner Mongolia, China

☯ These authors contributed equally to this work.
* siriguleng@imnu.edu.cn

## Abstract

Currently, pulmonary nodules detection work mostly focus on recognition and diagnosis of solid nodules. However, ground glass nodules have higher probability of malignancy, posing greater identification challenges and thus greater value for detection. To achieve rapid and accurate detection of ground glass nodules. This article proposed an algorithm based on RT-DETR model with the following enhancement: 1) optimize the backbone network with FCGE blocks to increase the detection accuracy of small-sized and blurred edge nodules; 2) replace the AIFI module with HiLo-AIFI module to reduce redundant computation and improve the detection accuracy of pure ground glass pulmonary nodules and mixed ground glass pulmonary nodules; 3) replace the DGAK module with CCFF module to address the issue of capturing complex features and recognition of irregularly shaped ground glass nodules. To obtain a more lightweight model, modules are designed for smaller number of parameters and higher computational efficiency. Model are tested on mixed dataset composed of LIDC-IDRI data and clinical data from cooperating hospitals. Compared to the baseline model, it shows an average precision improvement (mAP50/mAP50:95) of 2.1% and 1.7%, with a reduction parameters by 5.2 million. On a specialized dataset containing both pure and mixed ground glass nodules, our model outperformed the baseline model in all evaluation metrics. In general, the model proposed in this paper achieves improvement on lightweightness and detection accuracy. However, the model exhibits poor noise resistance and robustness, suggesting optimization in future work.

## Introduction

Lung cancer is the leading cause of cancer related death worldwide, posing a significant threat to human health [1]. Early detection of lung nodules can help preventing lung cancer before progressing to malignancy [2]. Based on density, lung nodules are classified into solid and ground glass types. Currently, most research on pulmonary nodule detection focus on solid nodules, and only a small group targeting ground-glass types. However, ground glass nodules have a higher likelihood of malignant transformation compared to solid ones [3]. It's also

**Data availability statement:** All relevant data are within the manuscript and its Supporting Information files.

**Funding:** This paper is supported by Inner Mongolia Natural Science Foundation (Grant No.2024LHMS06006); Inner Mongolia Health Commission Project(Grant No.202201395); Baotou Municipal Health Science and Technology Project(Grant No.wsjkkj2022120); Inner Mongolia College Students' Innovation and Entrepreneurship Training Program Projects (Grant No.s202410130004); Inner Mongolia Natural Science Foundation (Grant No.2024MS06008); Inner Mongolia Natural Science Foundation (Grant No.2022MS06002,2024LHMS06024); Scientific Research for the Public Hospitals of Inner Mongolia Academy of Medical Sciences (Grant No. 2023GLLH0211).There was no additional external funding received for this study.

more challenging to detect. Ground-glass nodules can be further categorized into pure ground glass nodules and mixed ground-glass nodules based on internal density. Pure ground glass nodules have relatively even density, while mixed ground glass nodules contain solid components in addition to the ground glass density. The complex density characteristics of these two types of ground glass nodules make it difficult for existing detecting approach to accurately distinguish them from normal tissue. Overall, compared to solid nodules, ground glass nodules not only have a higher probability of malignancy but also higher detection difficulties due to the unevenness in densities. Thus, there is significant value in conducting further research into detection methods specifically tailored for ground glass nodules.

Traditional methods of detecting lung nodules rely on physicians looking directly at CT scan images of the lungs to identify nodules, but there are significant limitations to this approach. As the demand for healthcare grows, physicians are under tremendous pressure to review many CT images in a short period, which not only adds to their workload but can also lead to missed or misdiagnosed tests in a state of fatigue. In addition, the number of CT slices in the lungs is so large and tightly arranged that relying on manual examination of each slice is inefficient and prolongs the diagnostic process, but may also delay the patient's valuable treatment time. Therefore, there is a need for new, more accurate and efficient methods to assist physicians in the detection of lung nodules. Deep learning models are networks with a multilayer structure that can automatically extract complex and abstract features from raw data, and after training the deep learning model, the model realizes the automatic recognition of the target through the learned data features. Deep learning has the advantage of high detection accuracy, speed and efficiency, and is particularly good at handling large amounts of data. Based on these advantages, deep learning models can be used to make up for the shortcomings of traditional lung nodule detection methods and realize method innovation in the field of lung nodule detection. Using deep learning models, we automatically realize lung nodule screening from dense lung slices based on automatically extracted lung nodule features, which improves detection speed and efficiency and reduces the burden on doctors. In addition, the deep learning model can dig deeper into the features of lung nodules in lung images, thus avoiding the occurrence of misdetection and missed detection. By utilizing deep learning models it is possible to enable more healthcare providers to perform lung nodule detection with limited resources, alleviating pain for patients and improving their cure rates.

Zhu et al. [4] proposed an end-to-end lung nodule detection model based on Hierarchical-Split HRNet and feature pyramid network with atrous convolution, which improved the performance for recognizing nodules in different shapes and sizes through multi-scale feature fusion and adversarial training. However, this approach comes with a high computational cost and slower model inference speed. Xiong et al. [5] introduced an algorithm based on model fusion and adaptive false positive reduction, which fuses nodule candidates proposed by two mighty detection network and designed an adaptive 3D convolutional neural network to reduce false positive rate. However, hospital equipment might not be able to meet the required computational capability. Ji et al. [6] improved upon YOLOv5 by incorporating enhanced CBAM modules and ASPP modules. This method achieves high detection speed but need further improvement when comes to complex or irregular shapes. Wu et al. [7] modified YOLOv7 by introducing small object detection layer, multi-scale receptive field module, and efficient omni-dimensional convolution, which improved performance for detecting small and irregularly shaped nodules. However, this method sacrifices detection speed.

Lung nodule detection can be classified as real-time object detection or end-to-end object detection methods. Real-time object detection can quickly capture targets in images within a short time. It features a lightweight network structure and high computational efficiency but has lower detection accuracy. End-to-end object detection completes the detection process through

a unified neural network model without the need for multi-stage processing. It has a complex and sophisticated network structure with lower computational efficiency but higher detection accuracy. The task of lung nodule detection emphasizes not only the accuracy of the model but also the detection speed, requiring the model to accurately detect lung nodules in a short time.

Therefore, this paper uses the Real-Time DEtection Transformer (RT-DETR) model [8] for lung nodule detection. The RT-DETR model is a real-time end-to-end object detector that combines the fast detection speed of real-time object detectors with the high detection accuracy of end-to-end object detectors. However, there are three challenges using the RT-DETR model for this task. Firstly, RT-DETR's computational requirements is too high, thus the model needs to be more lightweight. Secondly, ground glass nodules are characterized by blurry edges, small sizes, and irregular shapes. These characteristics significantly reduce the accuracy of the RT-DETR model. Lastly, Pure ground-glass pulmonary nodules and mixed ground-glass pulmonary nodules with indistinct borders with normal lung tissue make it difficult for the model to identify the actual edges, thus impacting pure ground-glass versus mixed pulmonary nodules accuracy. To address these three issues, this paper proposes improvements to the RT-DETR model to develop a lightweight model capable of accurately detecting ground glass nodules. The contributions of this paper are as follows:

1. optimized the backbone network with FCGE blocks, increasing computational efficiency and the detection accuracy for ground glass nodules with small sizes and blurry edges.

2. proposed HiLo-AIFI Module to improve upon the AIFI module, addressing the issue of computational redundancy and improving performance for different types of ground glass nodules.

3. Improved the CCFF module with DGAK Module to solve the problem of insufficient extraction of complex features and enhancing the detection accuracy for irregularly shaped ground glass nodules.

## Materials and methods

### Datasets

The dataset used in this paper consists of the publicly available LIDC-IDRI data [9] and a clinical data from a cooperating hospital (with a data privacy agreement signed). The LIDC-IDRI dataset was constructed by the National Cancer Institute (NCI) in collaboration with several healthcare organizations, and the collaborating hospitals' dataset is from THE SECOND AFFILIATED HOSPITAL OF BAOTOU MEDICAL COLLEGE. The date of access to the clinical dataset of the partner hospitals for this study is February 20th, 2024. In addition, the authors did not have access to individually participant-identifiable information during or after data collection, participants provided written informed consent, and minors were not included in this study. The ground glass nodules have been filtered out from the LIDC-IDRI dataset, so that all the lung nodules data used in this paper are ground glass nodules. The dataset contains a total of 837 lung CT slices, which, after data augmentation, amount to 1,252 slices. The data split are as follows:70% as the training set, 15% as the validation set, and 15% as the test set. Detailed information about the dataset is shown in Table 1.

### Data preprocessing

The paper uses a combination of threshold segmentation and morphological methods for lung parenchyma segmentation, distinguishing lung tissue from surrounding structures such as bones and muscles to reduce interference during detection. The lung parenchyma segmentation process is illustrated by Fig 1. Specifically, the workflow is

(1) Apply the OTSU algorithm on original lung CT scan to output binary image.

(2) Flood-fill the lung parenchyma with grayscale 255.

(3) Apply area opening operation to remove bubbles within the lung parenchyma that appeared after OTSU binarization, and perform a difference operation with the flood-filled lung parenchyma image.

(4) Apply area opening operation again to eliminate small connected regions in the image to obtain the lung parenchyma mask.

(5) Perform an AND operation to combine mask with the original CT image to obtain the lung parenchyma image.

Table 1. Information on public datasets and partner hospital datasets.

| Specification | LIDC-IDRI(ground glass lung nodules) | Clinical data from a cooperating hospital |
|---|---|---|
| CT slicer | 612 | 640 |
| size | ≥3mm与‹3mm | 3-30mm |
| CT resolution | 512 | 512 |
| Format | DICOM | DICOM |
| Annotation method | Annotated by 4 doctors independently and reviewed mutually | Annotated by 2 doctors independently and reviewed mutually |

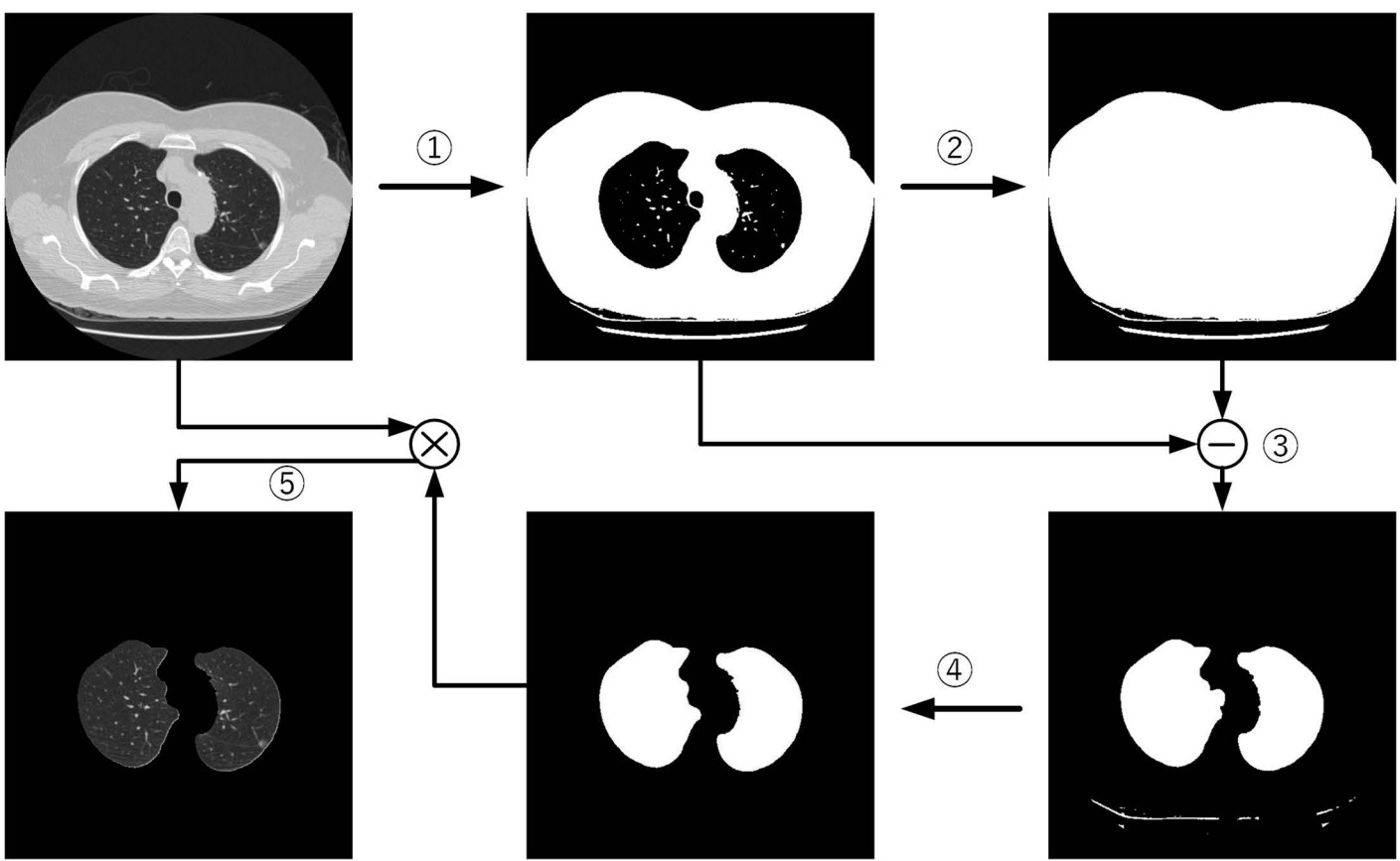

Fig 1. Lung parenchyma segmentation workflow.

## Evaluation indicators

The evaluation metrics used in this experiment include Precision, Recall, mean Average Precision (mAP50/mAP50-95), the number of model parameters (Params), and GFLOPs. Detailed information on the model evaluation metrics is shown in Table 2, where TP stands for True Positive, TN for True Negative, FP for False Positive, and FN for False Negative.

## Experimental setups and configurations

Experiments were conducted on the Ubuntu 18.03, the hardware configuration, software environment, and hyperparameter settings are as follows Table 3.

## RT-DETR model

RT-DETR is built upon the end-to-end object detector DETR [10], which addressed the issue of YOLO models' dependency on NMS (Non-Maximum Suppression). The NMS post-processing operation slows down the inference speed of YOLO models. However, the high computational cost of DETR prevented the no-NMS architecture from demonstrating speed advantage in real-time detection. Therefore, RT-DETR was designed as a new real-time end-to-end object detector based on DETR, whose detection speed and accuracy surpass those of YOLO models of the same scale.

The network structure of RT-DETR is shown in Fig 2, consisting of a backbone network, a hybrid encoder, and a decoder. In this paper we use ResNet18 as the backbone network, which has a compact structure and fewer parameters, enabling efficient feature extraction. The hybrid encoder effectively reduces computational work and improves inference speed by decoupling the interaction of Attention based Intra-scale Feature Interaction (AIFI) and the fusion of CNN based Cross-scale Feature Fusion (CCFF). AIFI employs a single-scale Transformer [11] encoder, performing feature interaction only on the highest-level feature

**Table 2. Detailed information on evaluation indicators.**

| Evaluation metrics | Definition | Math expression |
|---|---|---|
| Precision | Probability that the samples predicted to be positive are actually positive. | $Precision = \dfrac{TP}{TP+FP}$ |
| Recall | Proportion of actual positive samples that were correctly predicted as positive by the model. | $Recall = \dfrac{TP}{TP+FN}$ |
| mAP50/ mAP50-95 | Average precision of the model when using an Intersection over Union (IoU) threshold of 0.5/0.5-9.5. | $AP = \int_0^1 PRdr, mAP = \dfrac{1}{N}\sum_{i=0}^{n} APi$ |
| Params | Learnable weights and biases in the model, used to measure the model's size and complexity. A larger number of parameters indicates a more complex model. | – |
| GFLOPs | The number of floating-point operations per second. A higher GFLOPS value indicates a greater computational load of the model. | – |

**Table 3. Setups and hyperparameter settings.**

| Types | Configuration | Types | Value |
|---|---|---|---|
| GPU | NVIDIA L4 24GB | learning rate | 0.0001 |
| CPU | Intel(R)Xeon(R)Silver 4216CPU @ 2.10GHz | optimizer | AdamW |
| Python | 3.8.16 | epoch | 150 |
| Pytorch | 1.13.1 | batch | 4 |

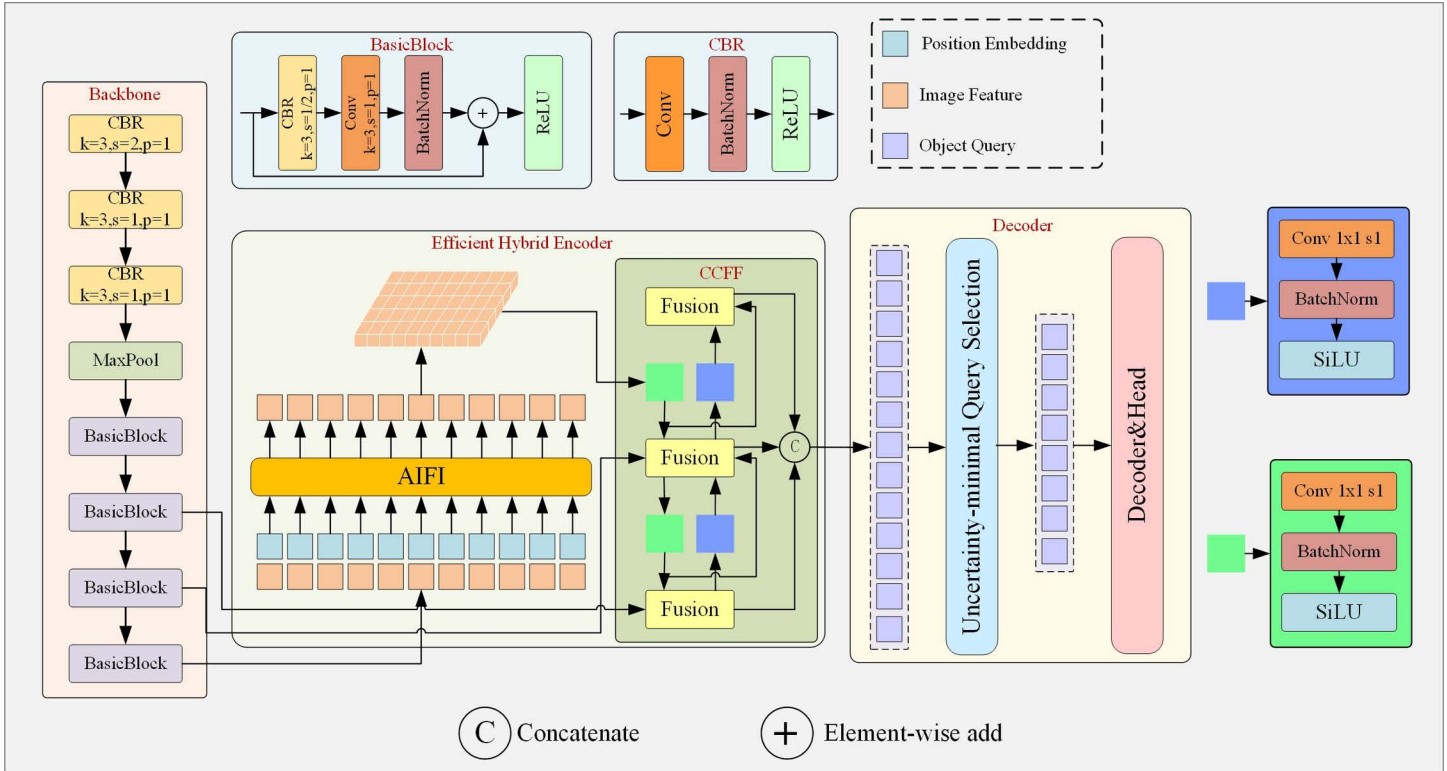

**Fig 2. Network structure of RT-DETR model.**

map which already contains information from lower-level features. CCFF is responsible for fusing features of different scales. It fuses the output features of AIFI with the last two features of the backbone network through multiple Fusion blocks. In the decoder, the quality of initial queries for the decoder is improved by explicitly optimizing the epistemic uncertainty. Specifically, uncertainty refers to the discrepancy between the predicted distributions of localization and classification. If the features have inconsistent predicted distributions for classification and localization, uncertainty arises. To optimized uncertainty, RT-DETR selected features with highly consistent distribution of classification and localization as initial queries.

## Network architecture for improved modeling

The workflow of this paper is shown in Fig 3. Initially, noises such as bone and blood vessels are removed from lung images. The preprocessed data is then input into an enhanced mode that optimizes the detection accuracy for small-sized, blurred edge ground glass nodules. Then the model is evaluated and validated.

For better performance, we optimized the model from two perspectives: 1) the improved model should be more lightweight with reduced number of parameters and improved computational efficiency to lower the computational requirement 2) the model structure is modified to address the issue of low detection accuracy caused by the characteristics of ground glass nodules. Structure of the improved RT-DETR model is illustrated by Fig 4. Detailed modification are as following. Firstly, to improve the computational efficiency and optimize the performance for small-sized and ill-defined nodules, we proposed replacing the BasicBlock in backbone network with FCGE-BasicBlock. Secondly, to address the redundancy of AIFI module, and improve the



**Fig 3. Workflow of this study.**

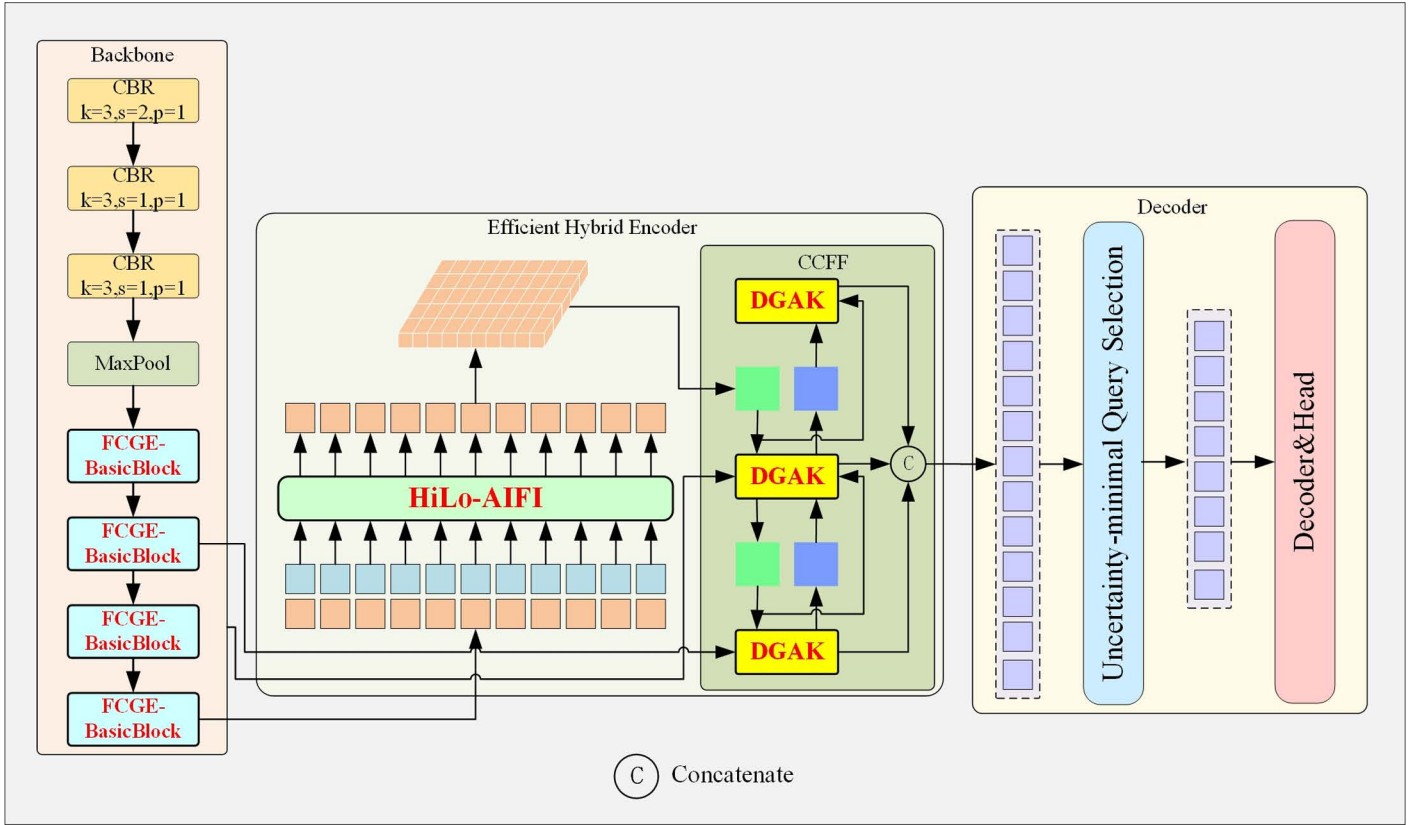

**Fig 4. Network structure of the improved RT-DETR model.**

accuracy on distinguishing pure ground glass nodules from mixed ground glass nodules, the AIFI module is replaced with HiLo-AIFI, where the Multi-head Self-attention (MSA) is updated to HiLo attention mechanism. Lastly, to enhance information exchange between features of different scales and improve the detection accuracy for irregularly shaped nodules, we improved the CCFF module by replacing the Fusion blocks in CCFF with the DGAK blocks. The modules proposed in this paper have two goals: the first is to make the model lightweight, and the second is to improve detection accuracy based on the characteristics of ground glass nodules. Through these improvements, the model achieves both a lightweight design and high detection accuracy.

## FCGE blocks

To make the model more lightweight, this paper proposes the FCGE blocks to reduce the number of parameters and increase computational efficiency. It can also help with detection

accuracy of small-sized and blurred-edge nodules. The FCGE block is an improvement based on FasterNet [12] whose structure is shown in Fig 5. It integrates the Efficient Local Attention (ELA) module [13] and Convolutional Gated Linear Unit (ConvGLU) module [14] into FasterNet. Specifically, the Partial Convolution (PConv) convolution enhances computational efficiency and reduces the number of parameters. The ELA attention module helps improve detection accuracy for small-sized nodules, and the ConvGLU module helps improve detection accuracy for blurred edge nodules. In this paper, the FCGE block is used to improve the backbone network ResNet18 by replacing the BasicBlock in ResNet18 with the FCGE-BasicBlock. Each BasicBlock in ResNet18 consists of two 3×3Conv layers with residual connections. Therefore, the specific improvement method in this paper is to replace the second 3×3Conv layer in the BasicBlock that has a stride of one with the FCGE Basic Blocks. Next we will elaborate on how the FCGE blocks achieved the improvements.

FasterNet has a simple structure and it is highly efficient in terms of performance and memory consumption without comprising its feature extraction capability and detection accuracy. These advantages makes it suitable for resource-constrained real-time tasks. FasterNet consists of one 3×3PConv layer and two 1× 1Conv layer, reusing input features through residual connections. 3×3PConv convolution is the core of the FasterNet, which only applies convolution operation to a portion of the input feature channels, while the rest channels remain unchanged. This helps reducing the computational load and memory access, without reducing model accuracy because the feature maps between different channels have a high degree of similarity. In other words, performing convolution operation only to part of channels will still be able to extract features effectively.

In this paper, the ConvGLU is used to replace the two 1× 1Conv layers in the FasterNet module. ConvGLU helps address the issue of ill-defined margin in targets. ConvGLU is a hybrid mechanism that combines Gated Linear Units (GLU) [15] and Depthwise Convolution (DWConv) to enhance information interaction between channels, as shown in Fig 6. Conv-GLU provides significant assistance in detecting targets with blurred edges through its unique gating mechanism and the combination with depthwise convolution. The gating mechanism

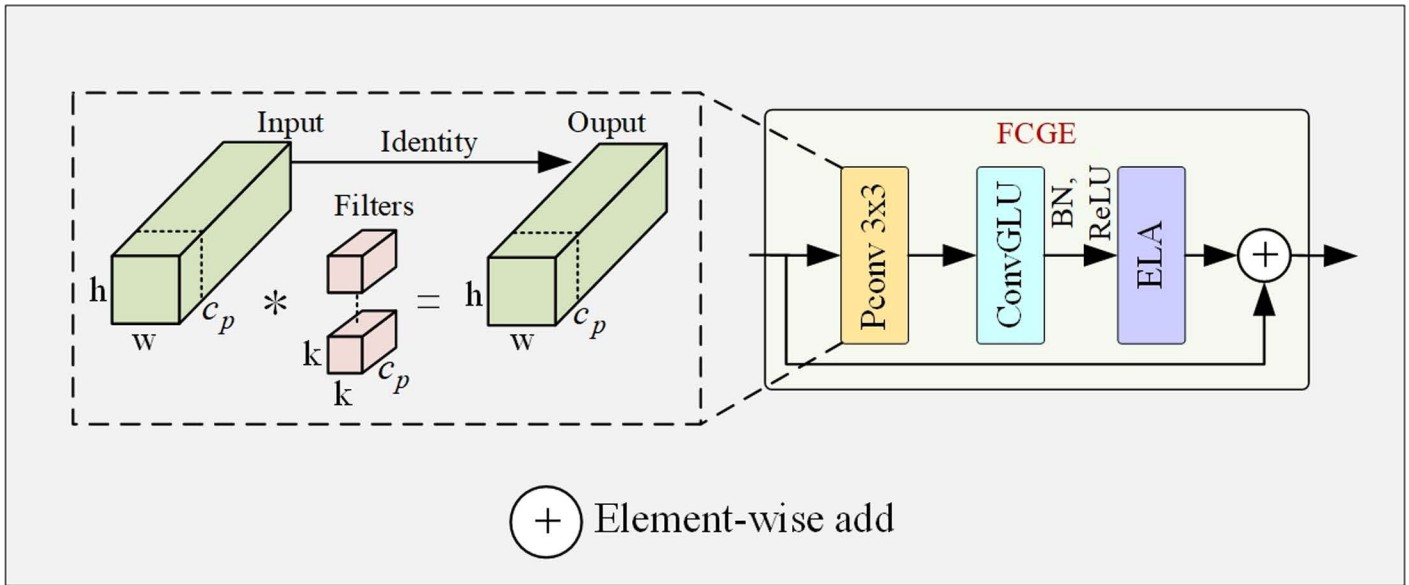

**Fig 5. Structure of FCGE module.**

of ConvGLU enables the model to dynamically adjust the response between channels, allowing the model to focus on critical features of the target even in cases of blurred edges. Through DWConv, ConvGLU can capture detailed features of local areas, including edge information, which is crucial for identifying targets with ill-defined edges. Thus, the integration of the ConvGLU module into the FCGE module enhances the ability to detect ground glass lung nodules with blurred edges.

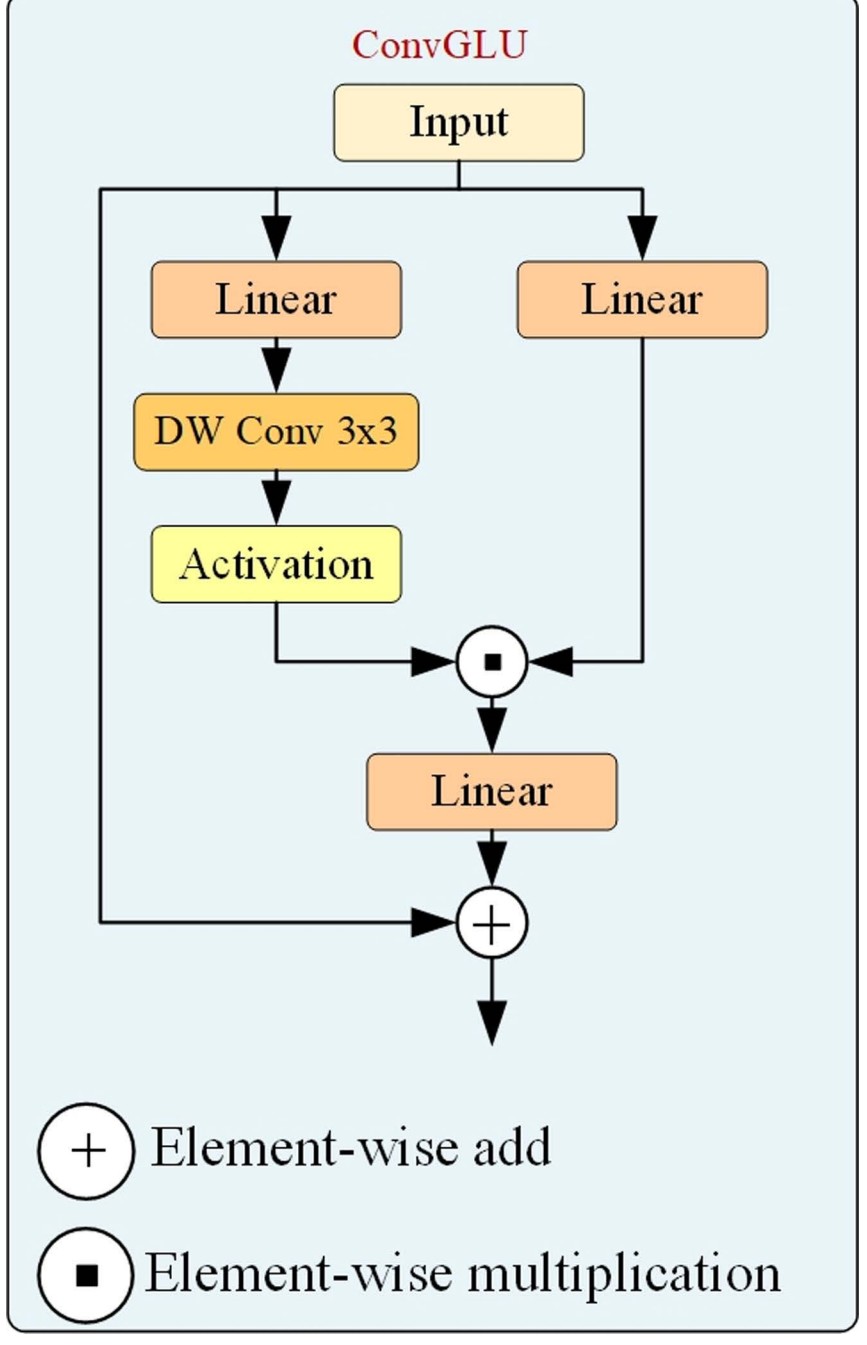

**Fig 6. ConvGLU module.**

For the detection of small-sized targets, both the channel and spatial dimensions are important. The channel dimension captures rich feature information, while the spatial dimension accurately locates the position of small targets, ensuring detection accuracy. Currently, effective utilization of the spatial dimension is challenging which usually comes at the cost of reduced channel dimensions or complicated network structure. To address this issue, we use the ELA module, whose structure is shown in Fig 7. First, strip pooling is used to capture

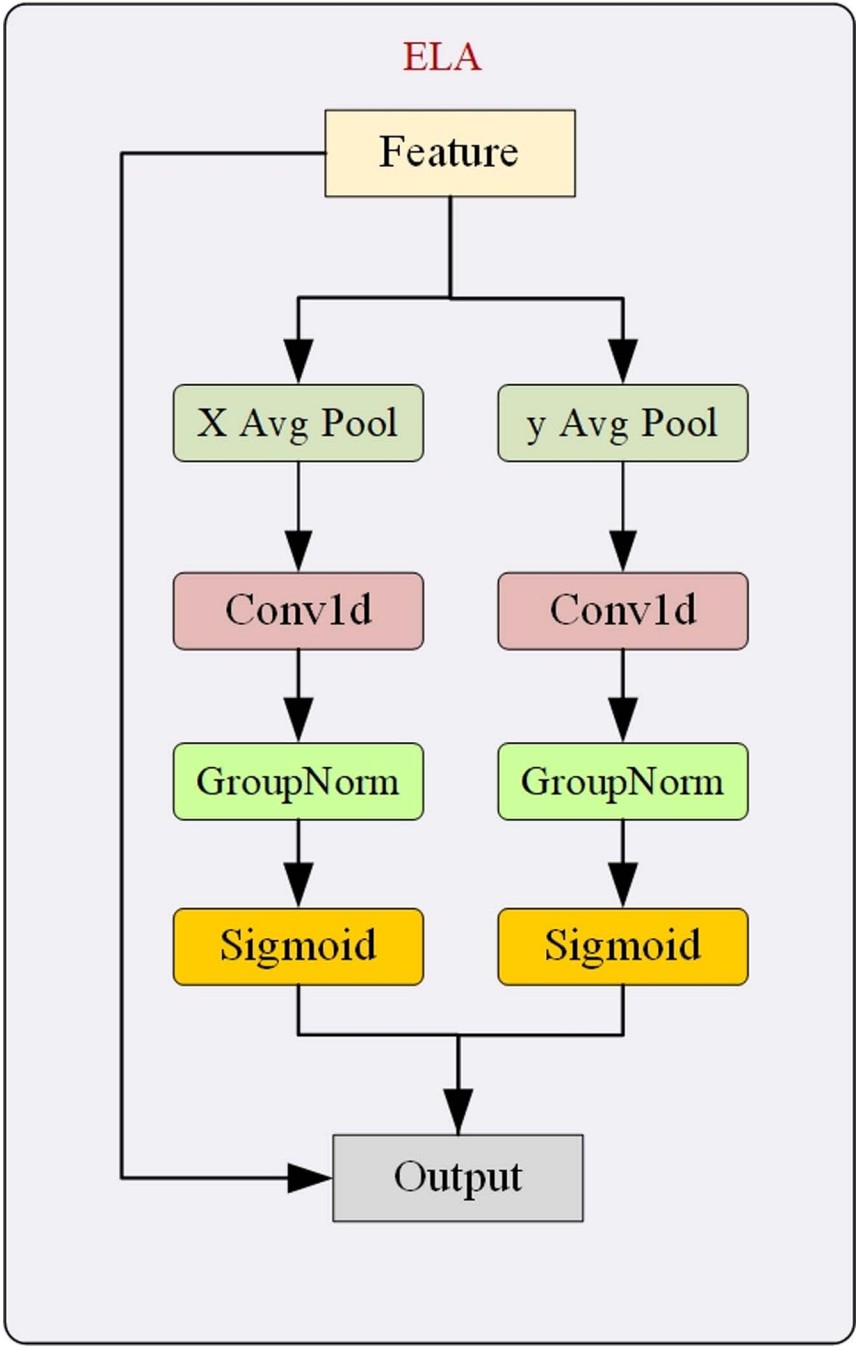

**Fig 7. ELA module.**

long-range dependencies in the spatial dimension. Then, by applying one-dimensional convolutions in different directions, spatial position information is effectively encoded, increasing sensitivity to spatial features. Finally, group normalization is performed for model stability and better handling of spatial features. Introduction of the ELA module significantly enhanced the FCGE module's ability to detect small-sized ground glass lung nodules.

The innovation of the FCGE module is its deep integration with the ConvGLU module and ELA attention module on the basis of the FasterNet module, which not only greatly improves the computational efficiency and effectively reduces the parameter scale, but also shows excellent performance enhancement in detecting tiny and fuzzy-edge lung nodules. FCGE modules accurately address multiple challenges with a one-stop solution, realizing a double leap in computational efficiency and detection accuracy. Ultimately, this paper combines the Basic-Block module in the backbone network with the FCGE module to improve the performance of the model as a whole.

## HiLo-AIFI module

The AIFI module is the core of RT-DETR for achieving intra-scale feature interaction. AIFI is essentially a Transformer encoder, primarily capturing local area features through MSA. MSA distributes the input sequence across multiple sub-attention heads and calculates attention maps independently. This approach is very effective in improving performance, but it causes computational redundancy. Although multiple attention heads are designed to capture different aspects of the input data, many heads actually learn similar feature representations, resulting in redundant information during computation and not improving model performance. Additionally, AIFI module works the best for the detection accuracy for both pure ground glass lung nodules and mixed lung nodules. Because it focuses on capturing detailed information within local areas, which is crucial for identifying subtle density changes within lung nodules. While MSA can capture global dependencies, it is not sensitive enough for expressing local fine features. Therefore, to address the problem of MSA computational redundancy and Therefore, to address the problem of MSA computational redundancy and Improving the accuracy of detection of pure ground-glass nodules and mixed ground-glass nodules, this paper proposes the HiLo-AIFI module [16], which replaces the MSA attention module in the AIFI module with the HiLo attention module.

The HiLo attention module is shown in Fig 8. It divides the MSA layer into two paths where one path encodes high-frequency interactions via local self-attention while the other path encodes low-frequency interactions via global attention. The total amount of heads are divided into two groups, (1-α) N_hheads to High Frequency Attention(Hi-Fi) group and αN_hheads to Lo-Fi group, where N_his the total amount of heads, α is the split ratio. Hi-Fi captures fine-grained high-frequency features through local window self-attention, which contains local details of the target. Low Frequency Attention (Lo-Fi) uses down-sampled features from average pooling as input and captures low-frequency features through global attention, which relates to the global structure and smooth regions of the image. This separation method allows the model to focus on features that are more critical to the task rather than evenly distributing computational resources across all features, thereby improving computational efficiency. HiLo assigns features to different heads, with each head focusing on specific input features, reducing redundant computation between heads. By addressing the computational redundancy of MSA, it improves the overall computational efficiency of the model.

The density of pure ground glass lung nodules increases uniformly, which does not obscure the vascular textures in the lungs. This results in an indistinct boundary with surrounding normal tissue, leading to lower detection accuracy. The density of mixed ground glass lung nodules increases inconsistently, which can cause difficulties for the model when extracting

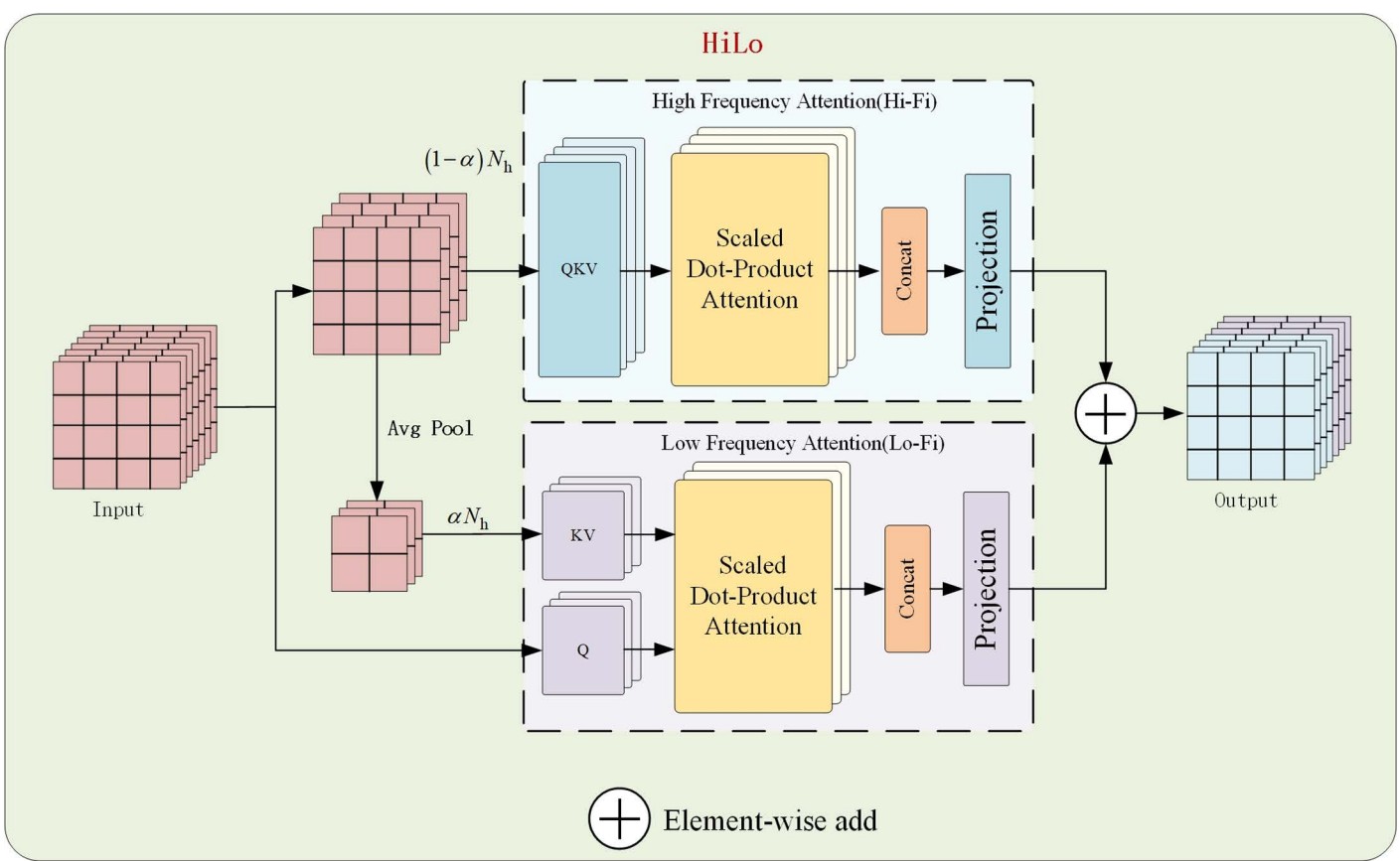

**Fig 8. HiLo-AIFI module.**

key features, as they may vary across regions of different densities, also leading to lower detection accuracy. The Hi-Fi path focuses on high-frequency regions, which contain a lot of detailed information. It enhances the expression of local features through a local window self-attention mechanism, capturing internal texture details of lung nodules and the uniformity of density, which helps recognizing differences in density. Lo-Fi provides information on whether there are large areas of density variation within the nodule through global attention, assessing the overall density of the nodule. By using the HiLo attention module, not only is the computational redundancy of MSA addressed, improving the model's computational efficiency, but it also effectively enhances the detection accuracy for both pure and mixed ground glass lung nodules.

The innovation of this module is that this paper uses the HiLo module to replace the MSA module in the AIFI module, effectively combines the HiLo module with the AIFI module, and creates a new HiLo-AIFI module, which solves the problems of computational redundancy and difficulty in capturing the local feature information in the original AIFI module, sufficiently improves the overall computational efficiency of the model, and effectively strengthens the ability to extract and express local features, helping the model to improve the detection accuracy of pure ground-glass and mixed ground-glass lung nodules. It can improve the overall computational efficiency of the model, effectively enhance the local feature extraction and expression ability, and help the model improve the detection accuracy of pure ground-glass and mixed ground-glass lung nodules.

## DGAK module

The Fusion module is crucial in the RT-DETR model for cross-scale feature fusion. The network structure of the Fusion module is shown in Fig 9. It consists of two 1×1Conv layers to adjust the number of channels, and N RepBlocks composed of RepConv [17] for feature fusion. The output from two paths is fused through element-wise addition. For the task of detecting ground glass lung nodules, the Fusion module has three drawbacks. The first drawback is that this fusion method is relatively simple and cannot fully capture complex characteristics of the input data. The second drawback is that the stacking of RepConv layers in the Fusion module may result in a high computational load. Lastly, as ground glass nodules have irregular shapes, and the features presented on feature maps at different scales vary, causing information loss or feature confusion, thereby affecting recognition accuracy. The Fusion module is less useful for improving the accuracy of detecting ground-glass lung nodules with irregular morphology. To address the shortcomings mentioned above, this paper proposes the DGAK module as a replacement for the Fusion module. The DGAK module enhances information interaction between features at different scales, fully captures the complex characteristics of input data, and improves computational efficiency during fusion. Most importantly, DGAK module can effectively improve the detection accuracy of irregularly shaped ground glass nodules. The structure of the DGAK module is shown in Fig 10. The DGAK module is an improved version of the Dynamic Grouped Convolution Shuffle Transformer (DGST) module [18], where the grouped convolution in the DGST module is replaced with Alterable Kernel Convolution (AKConv) [19]. The DGST module has the capability to enhance information exchange between features and improve computational efficiency during fusion, while the AKConv can effectively extract features of irregularly shaped ground glass nodules. Replacing the Fusion module with the DGAK module helps improve the overall computational efficiency of the model and the detection accuracy of irregularly shaped nodules. The following section will explain how the DGAK module addresses the three shortcomings of the Fusion module.

The DGST module has streamlined network structure, minimized model parameters, and improved computational efficiency. The core of the DGST module is a 3:1 division strategy, where the portion with a ratio of 3 goes through a combination of grouped convolution and channel shuffle operation. Grouped convolution reduces the number of model parameters by grouping the input channels and performing operations on each group separately. Channel shuffle is a network optimization technique that promotes information flow by rearranging and mixing channels from different groups, facilitating interaction between features and more

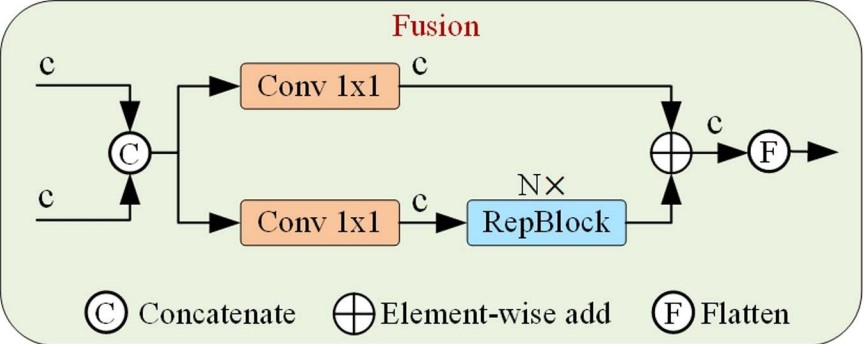

**Fig 9. Structure of Fusion module.**

comprehensive feature fusion. The portion with a ratio of 1 is directly fed into the convolutional feed-forward network (ConvFFN). Here, ConvFFN replaces the fully connected layers, helps with reducing the number of parameters while maintaining the dimensionality of the feature space.

AKConv is a new convolution operation that overcomes the limitations of traditional ones. Its convolution kernels adopt non-fixed sampling shapes, enabling it to adapt to irregular and diverse target shapes. The core of AKConv are the initial sampling coordinate generation algorithm and the adjustable offset mechanism. In traditional convolution operation, both kernel shape and its sampling grid is fixed. In contrast, the shape of AKConv's convolution kernel is irregular. To ensure sampling effectively, a new algorithm was developed to generate initial sampling coordinates for convolution kernels of any size. An illustrative diagram of the

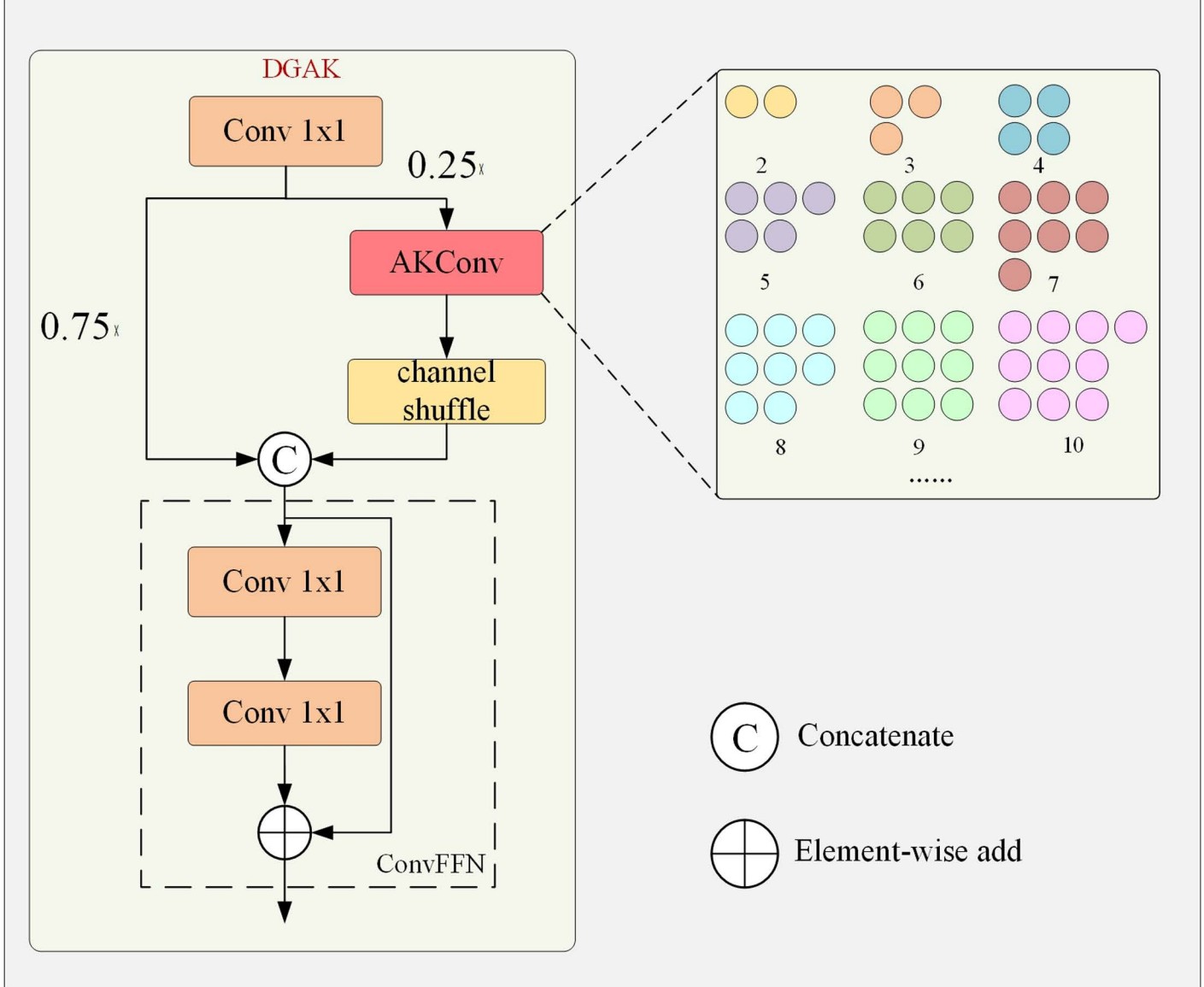

**Fig 10. Architecture of the DGAK Module and AKConv Initial Sampling Coordinate Generation Algorithm.**

algorithm is shown in Fig 10. The algorithm works as follows: first, a basic regular sampling grid is created. Then, for the remaining parts, the algorithm generates additional irregular sampling points. Finally, the basic regular sampling grid and the irregular sampling grid are stitched together to form an overall sampling grid, which includes all the necessary sampling points for the convolution kernel on the input feature map. The offset mechanism of AKConv allows the convolution kernel to dynamically adjust the sampling positions to adapt to the shape and size of the target. The specific method is as follows: first, the corresponding offset is calculated through a convolution layer. Then, the offset is combined with the original sampling coordinates of the convolution kernel to adjust the actual position of each sampling point. Finally, using the adjusted sampling coordinates, AKConv performs interpolation and resampling to obtain the features at the corresponding positions. This paper replaces the grouped convolution in the DGST module with AKConv and combines that with channel shuffle. AKConv helps adapting to targets in different sizes and shapes, extracting rich features. Channel shuffle enhances the interaction between features. Combination of the two optimizes computational resources and improves the detection accuracy of irregularly shaped ground glass nodules.

The innovation of this module is to replace the grouped convolution in the DGST module with the AKConv convolution and create a new DGAK module through the deep fusion of the DGST module and AKConv, which enhances the ability of information interaction between features of different scales and achieves the ability to optimize the network structure, reduce the parameters of the model, and efficiently extract the features of the morphologically irregular ground-glass-type lung nodules. In this paper, the original Fusion module in the RT-DETR model was replaced with the DGAK module to help the model improve the accuracy of detection of morphologically irregular ground-glass lung nodules.

## Results

### Ablation study

To verify that the proposed improvement module has a lightweight effect while maintaining detection accuracy, we designed an ablation study. The dataset and experimental environment used are the same. As the result shown in Table 4, the improved modules can reduce the model parameters by 5.2 million at least, and can improve computational efficiency. Usually, lightweight improvements to a model compromises detection accuracy. However, in this study the targeted improvements to each module not only avoid sacrificing detection accuracy but also slightly enhance precision and mean average precision (mAP). The precision improved by up to 2.4%, and the mean average precision (mAP50/mAP50:95) increased by up to 2.1% and 1.7%, respectively.

**Table 4. Ablation study results.**

| FCGE | HiLo-AIFI | DGAK | Precision(%) | Recall(%) | mAP50(%) | mAP50:95(%) | Params(M) | GFLOPs(G) |
|------|-----------|------|--------------|-----------|----------|-------------|-----------|-----------|
| — | — | — | 93.3 | 88.7 | 92.8 | 45.0 | 19.9 | 56.9 |
| √ | — | — | 94.2 | 89.6 | 93.7 | 45.4 | 17.1 | 48.3 |
| — | √ | — | 94.4 | 89.9 | 93.6 | 45.8 | 18.7 | 49.0 |
| — | — | √ | 94.7 | 89.7 | 93.8 | 45.6 | 18.6 | 50.8 |
| √ | √ | — | 95.0 | 90.6 | 94.2 | 46.1 | 16.0 | 46.4 |
| √ | √ | √ | 95.7 | 91.0 | 94.9 | 46.7 | 14.7 | 40.2 |

## Comparison with SOTA models

To verify that the improved model proposed model has improved computational efficiency and detection accuracy, comparisons with current mainstream object detection models are made. The dataset and experimental environment are the same. The baseline used in this paper is the RT-DETR model with ResNet18 as the backbone network, model size corresponds to the size L in the YOLO series. According to the comparison results shown in Table 5, although the baseline model used in this paper has the fewest parameters and the best computational efficiency, lightweighting of the model is still necessary to better adapt to resource-constrained devices while maintaining high accuracy. Results indicates that the RT-DETR series generally has less parameters and computational complexity compared to the YOLO series, and the detection accuracy is also higher than that of the YOLO series. Moreover, to obtain desired results, the YOLO series requires more epochs than the RT-DETR series. Additionally, there is a significant variation in detection performance among versions of the YOLO series. For example, YOLOv5 [20] and YOLOv8 [21] perform well, while other versions perform relatively poorly. In contrast, the performances variation among RT-DETR series is relatively small.

Depending on the backbone network, RT-DETR model has different versions: RT-DETR-L uses HGNetv2 as the backbone network, while RT-DETR-R18/R34/R50 use ResNet18/34/50 as the backbone network. Among those versions, RT-DETR-R18 has the fewest parameters and computational complexity, but its detection accuracy, recall, and mean average precision are not as high as others. After enhancement proposed in this paper, the RT-DETR-R18 model is not only further lightweighted, but also shows slight enhancements in detection accuracy, recall, and mean average precision compared to other versions. Compared to YOLO models, the model in this paper shows higher computational efficiency and better performance in other evaluation metrics such as detection accuracy. Overall, the improved model outperforms both RT-DETR and YOLO series in this study.

The end-to-end detection model and the two-stage detection model have higher detection accuracy and average accuracy. In this paper, the mainstream end-to-end detection model Swin Transformer [22] and the two-stage detection model Faster R-CNN [23] are tested in the

**Table 5. Comparison of object detection models.**

| Model | Precision(%) | Recall(%) | mAP50(%) | mAP50:95(%) | Params(M) | GFLOPs(G) |
|---|---|---|---|---|---|---|
| Swin Transformer | 94.2 | 90.1 | 93.8 | 45.8 | 48.5 | 135.2 |
| Faster R-CNN | 93.2 | 89.8 | 93.1 | 45.6 | 48.0 | 120.1 |
| SSD | 87.1 | 84.1 | 85.3 | 40.4 | 41.2 | 113.4 |
| Chostnetv2 | 90.1 | 88.2 | 92.3 | 43.1 | 42.5 | 119.6 |
| YOLOv5-L | 88.2 | 85.6 | 91.4 | 42.1 | 43.6 | 126.3 |
| YOLOv6-L | 87.3 | 86.6 | 91.8 | 43.7 | 47.8 | 131.2 |
| YOLOv7-L | 85.3 | 73.8 | 80.3 | 40.4 | 36.4 | 118.2 |
| YOLOv8-L | 88.6 | 87.7 | 91.6 | 42.4 | 43.1 | 124.7 |
| YOLOv9 | 91.7 | 88.3 | 92.8 | 43.9 | 50.3 | 136.8 |
| YOLOv10 | 87.8 | 85.2 | 90.5 | 41.7 | 31.7 | 106.3 |
| RT-DETR-L | 93.6 | 86.6 | 93.8 | 45.4 | 30.9 | 103.4 |
| RT-DETR-R18 | 93.3 | 88.7 | 92.8 | 45.0 | 19.9 | 56.9 |
| RT-DETR-R34 | 93.8 | 90.1 | 93.6 | 45.7 | 31.6 | 121.8 |
| RT-DETR-R50 | 94.8 | 90.3 | 94.1 | 46.1 | 51.9 | 139.5 |
| Our | 95.7 | 91.0 | 94.9 | 46.7 | 14.7 | 40.2 |

same experimental environment and dataset, and the results in Table 5 show that the detection accuracy and the average accuracy of these two detection models are lower than that of the improved model in this paper, and the detection speed is also slower than the improved model in this paper and also has a larger number of parameters. The real-time detection model has a faster detection speed with a smaller number of parameters, and this paper similarly compares the classical real-time target detector SSD [24] experimentally with Chostnetv2 [25], which is based on a lighter and improved version of the YOLO model with a faster detection speed. According to the experimental results in Table 5, it can be seen that the detection speed of these two detection models is slower than that of the improved model in this paper and the number of parameters is similar to that of the improved model in this paper, and the detection accuracy and average accuracy are much lower than that of the improved model in this paper. By comparing with the classical target detection model, the improved model in this paper still has a big advantage.

## Experiments on specialized datasets

To demonstrate that the improvements made in this paper effectively enhance the detection accuracy of pure ground glass nodules and mixed ground glass nodules, we conducted experiments on specialized data. The dataset used for experiments mentioned earlier contains a mix of pure and mixed ground glass nodules. For this experiment, we constructed two specialized lung nodule datasets: one consists of only pure ground glass nodules and the other only mixed ground glass nodules. Comparison of the improved model and the baseline model were conducted on these two specialized datasets. Each of these specialized datasets contains 200 CT slices. The setup and environment remained the same. The baseline model, as well as YOLOv5 and YOLOv8 (which has leading performance among the YOLO series), and RT-DETR-R34 and RT-DETR-R50 (which has leading performance among the RT-DETR series), were compared.

As shown in Table 6, when testing on pure ground glass nodules datasets, our model achieves the best detection performance, with the best result across all evaluation metrics. Table 7 indicates that when testing on mixed ground glass nodules dataset, the proposed model also outperforms others. Comparing results in Table 6 and Table 7, we found that the performance on detecting pure ground glass nodules is overall better than on mixed ground glass nodules, indicating that pure ground glass nodules are easier to detect. Meanwhile, due to the significantly reduced complexity of the datasets, the detection results in this experiment are better than the previous results obtained on the mixed dataset. These experimental results demonstrate that the improvements made to the model in this paper are effective, as it achieves high detection accuracy for both pure ground glass nodules and mixed ground glass nodules.

**Table 6. Model comparison on pure ground glass lung nodules dataset.**

| Model | Precision(%) | Recall(%) | mAP50(%) | mAP50:95(%) |
|---|---|---|---|---|
| YOLOv5-L | 90.1 | 86.5 | 92.3 | 43.4 |
| YOLOv8-L | 90.7 | 88.3 | 92.7 | 43.8 |
| RT-DETR-R18 | 94.5 | 90.1 | 93.7 | 45.8 |
| RT-DETR-R34 | 94.9 | 90.6 | 94.3 | 46.1 |
| RT-DETR-R50 | 95.6 | 91.4 | 94.8 | 47.4 |
| Our | 96.4 | 92.3 | 95.8 | 48.1 |

**Table 7. Model comparison on mixed ground glass lung nodules dataset.**

| Model | Precision(%) | Recall(%) | mAP50(%) | mAP50:95(%) |
|---|---|---|---|---|
| YOLOv5-L | 89.5 | 86.0 | 91.7 | 42.6 |
| YOLOv8-L | 90.3 | 87.7 | 92.2 | 43.4 |
| RT-DETR-R18 | 93.8 | 89.2 | 93.2 | 45.3 |
| RT-DETR-R34 | 94.1 | 90.2 | 93.9 | 45.6 |
| RT-DETR-R50 | 95.0 | 90.5 | 94.3 | 46.6 |
| Our | 95.9 | 91.4 | 95.2 | 47.3 |

## Visualization analysis experiment

To more explicitly verify the improved accuracy and inference speed of the model proposed in this paper, a visualization analysis experiment is conducted. The improved model is used to detect mixed, pure ground glass nodules, small-sized nodules, edge-blurred nodules, and irregularly shaped ground glass nodules. Additionally, to demonstrate the effectiveness of the model improvements, YOLOv5 and YOLOv8, RT-DETR-R34 and RT-DETR-R50, were used as comparison models. As shown in Fig 11, the highlighted parts in the heatmap correspond to the actual positions of the ground glass nodules as outlined in the GroundTruth. Because detecting mixed, small-sized, edge-blurred, and irregularly shaped ground glass nodules is more challenging. When detecting these types of nodules, the performance of the YOLO series models is significantly poorer, while the RT-DETR series models relatively better. Compared to the baseline model, our model shows enhancements in both detection accuracy and inference speed. Although the detection accuracy of our model is occasionally on par with RT-DETR-50, the inference speed is noticeably faster. For pure ground glass nodules, which are relatively easier to detect, the performance gaps between the YOLO models and the RT-DETR models are small, and our model still achieves the highest detection accuracy and the fastest inference speed. The visualization analysis experiment results further verify that the improvements made to the model are effective, enabling precise detection of different types of ground glass nodules at a faster inference speed.

## Limitations

Although experimental results above shows that the improved model in this paper has the advantages of fast inference speed and high detection accuracy, it also has some drawbacks, such as poor noise resistance and weak robustness. In the aforementioned experiments, the datasets used were preprocessed to remove noise interference from bones, trachea, and other elements in the images. We test the improved model using an unprocessed dataset and compare the results with those obtained from the preprocessed dataset, for a detailed analysis of the model's weaknesses in noise resistance and robustness. As shown in Table 8, model performance on unprocessed dataset is significantly pooper compared to performance on preprocessed dataset. Fig 12 shows that the detection accuracy of the improved model on the unprocessed dataset dramatically declines compared to the preprocessed dataset. Further, for some more challenging detection tasks, such as inferring mixed ground glass nodules or small-sized ground glass nodules, the model exhibits missed and false positive cases. This means our model has weak noise resistance and robustness, as significant noise in the dataset severely impacts the detection accuracy of the model.

In the future work, we will rely on large model to enhance the model's noise resistance and improve its robustness. Large models have the ability to process multiple data types, analyzing text and image simultaneously and build connections between them. Linking clinical medical

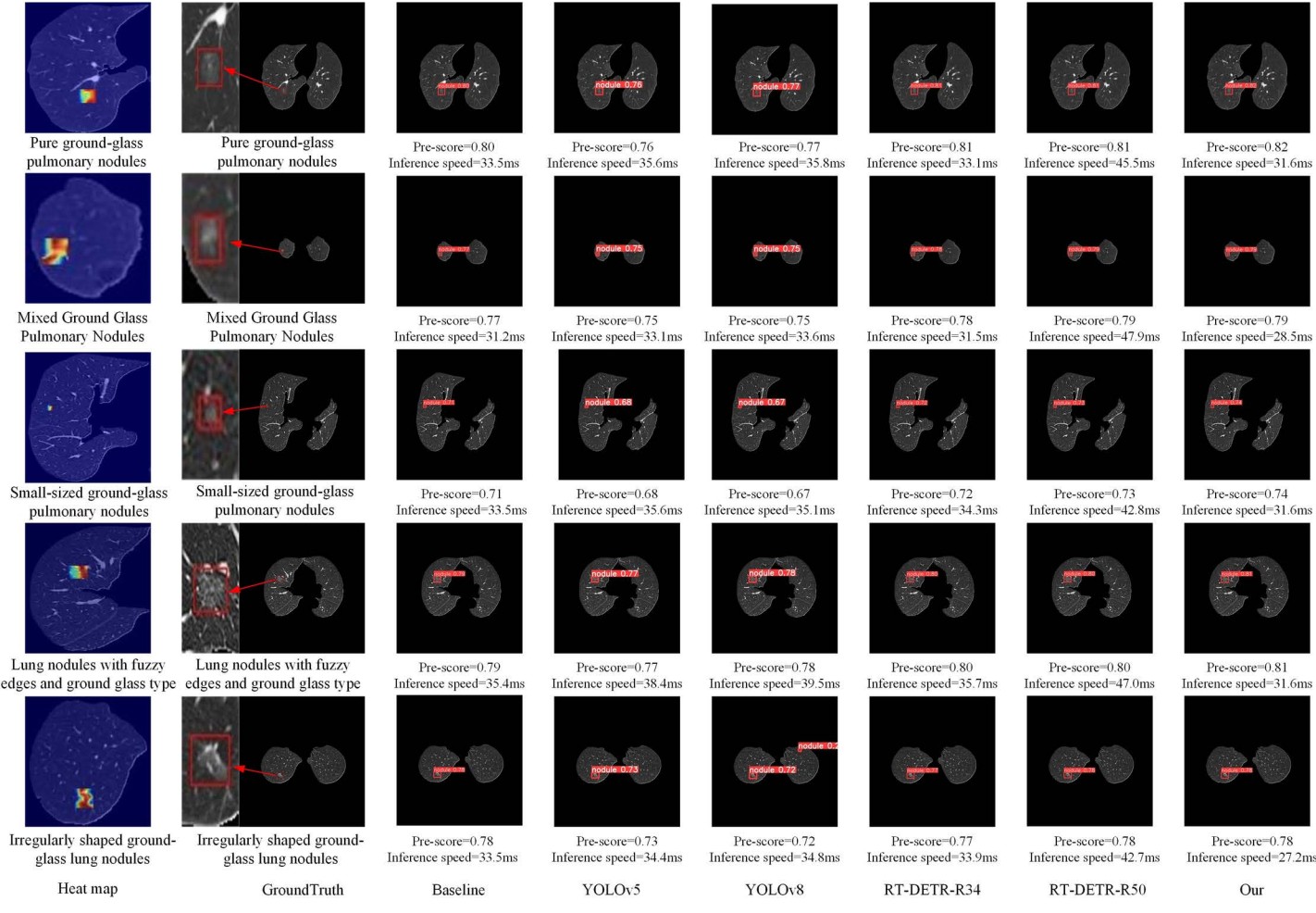

**Fig 11. Results of Visualization Analysis Experiment.**

**Table 8. Test result on unpreprocess data and preprocess data.**

| Preprocessed or Not | Precision(%) | Recall(%) | mAP50(%) | mAP50:95(%) |
|---|---|---|---|---|
| Yes | 95.7 | 91.0 | 94.9 | 46.7 |
| No | 78.6 | 76.4 | 79.5 | 33.1 |

reports with lung CT scans via large model, it will be less susceptible to image noise, leading to more accurate detection results.

## Conclusions

To quickly and accurately detect ground glass lung nodules, this paper proposes an improved RT-DETR model with the following enhancement. First, to increase the detection accuracy for nodules with blurred edges and small sizes, this paper introduces the FCGE module to optimize the backbone network. Second, to address the computational redundancy of the AIFI module, this paper proposes the HiLo-AIFI module as a replacement for the AIFI module. The HiLo-AIFI module also helps with the detection accuracy of both pure and

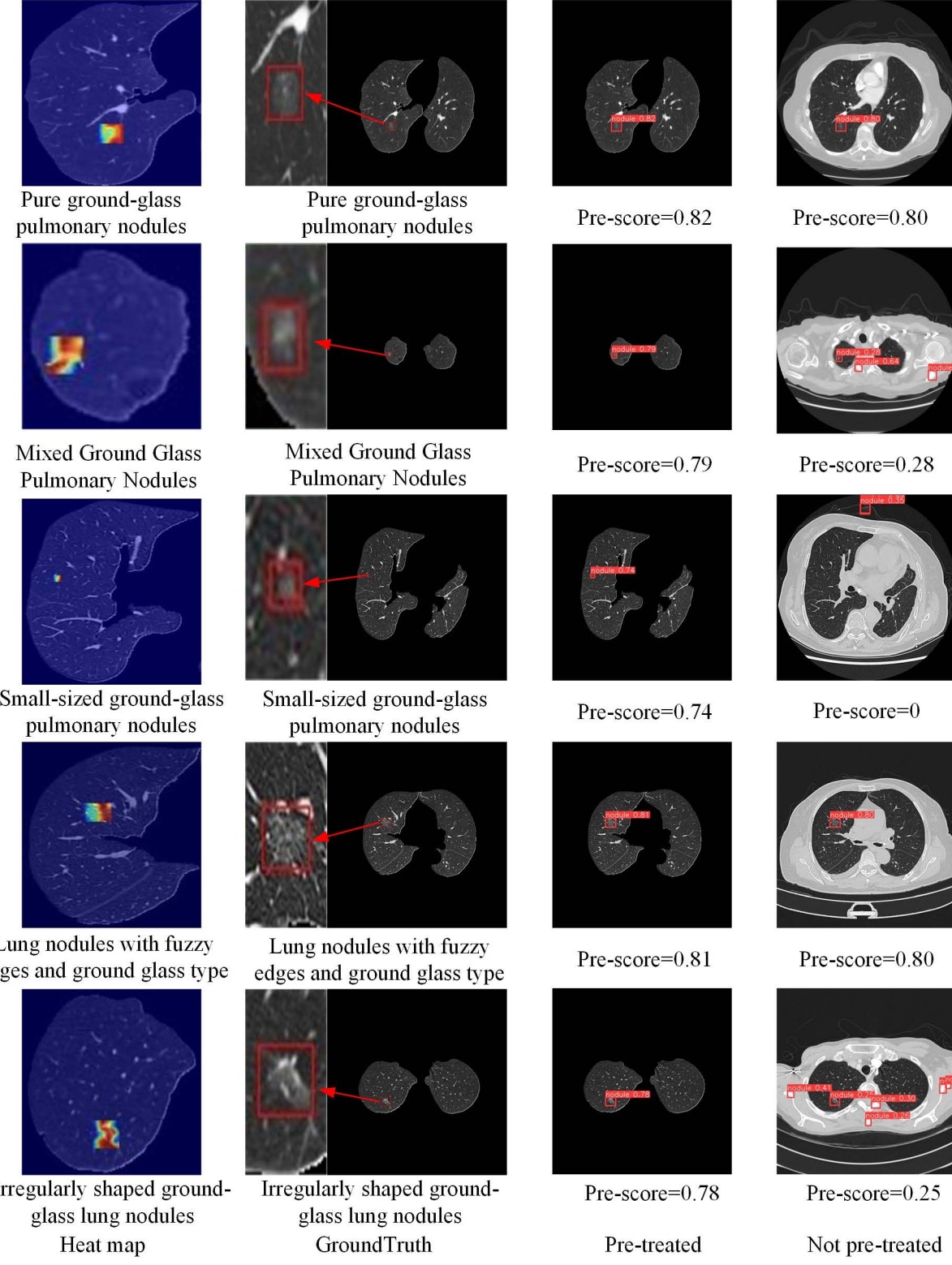

**Fig 12. Visualization of the drawbacks of the improved model.**

mixed ground glass nodules. Finally, to solve the issue of the model's inability to fully extract complex features and to improve the detection accuracy of irregularly shaped nodules, this paper introduces the DGAK module and enhance the CCFF module. To achieve a lightweight model, all the proposed modules have an effect of reducing model parameters and improving computational efficiency. The improved model was tested against the baseline model on a mixed dataset from LIDC-IDRI and clinical data from cooperating hospitals. With the number of parameters decreased by 5.2 million, the model precision increased by 2.4%, the mean average precision (mAP50/mAP50:95) increased by 2.1% and 1.7% respectively. Overall, compared to other object detection models, the improved model demonstrated the highest computational efficiency and detection accuracy. When tested on specialized datasets containing pure ground glass nodule or mixed ground glass nodule, the improved model achieved the best performance across all evaluation metrics. The limitation of this paper's model is its poor noise resistance and weak robustness. When there is significant noise, antifacts, or anomalies in the lung CT scans, the model's detection accuracy significantly decreases. In future work, we plan to leverage large models to strength the noise resistance and robustness.

## Author contributions

**Conceptualization:** Siyuan Tang, Qiangqiang Bao.

**Data curation:** Qingyu Ji.

**Formal analysis:** Tong Wang, Jinliang Zhao, Yuhan Qu.

**Investigation:** Tong Wang, Naiyu Wang.

**Methodology:** Siyuan Tang, Qiangqiang Bao.

**Project administration:** Siriguleng Wang.

**Software:** Qiangqiang Bao.

**Validation:** Min Yang, Yu Gu.

**Writing – original draft:** Siyuan Tang, Qiangqiang Bao.

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
