## [Decision Letter · Decision Letter 0]

20 Nov 2024

PONE-D-24-35347Improvement of RT-DETR model for ground glass pulmonary nodule detectionPLOS ONE

Dear Dr. Wang,

Thank you for submitting your manuscript to PLOS ONE. After careful consideration, we feel that it has merit but does not fully meet PLOS ONE’s publication criteria as it currently stands. Therefore, we invite you to submit a revised version of the manuscript that addresses the points raised during the review process.

We look forward to receiving your revised manuscript.

Kind regards,

Kannadhasan Suriyan

Academic Editor

PLOS ONE

**Journal Requirements:**

This paper is supported by Inner Mongolia Health Commission Project(Grant No.202201395); Baotou Municipal Health Science and Technology Project(Grant No.wsjkkj2022120); Inner Mongolia College Students' Innovation and Entrepreneurship Training Program Projects(Grant No.s202310130004); Inner Mongolia College Students' Innovation and Entrepreneurship Training Program Projects (Grant No.s202410130004); Inner Mongolia Natural Science Foundation (Grant No.2024LHMS06006); Inner Mongolia Natural Science Foundation (Grant No.2024MS06008); Inner Mongolia Natural Science Foundation (Grant No.2021MS06026); Scientific Research for the Public Hospitals of Inner Mongolia Academy of Medical Sciences (Grant No. 2023GLLH0211).

This paper is supported by Inner Mongolia Health Commission Project(Grant No.202201395); Baotou Municipal Health Science and Technology Project(Grant No.wsjkkj2022120); Inner Mongolia College Students' Innovation and Entrepreneurship Training Program Projects(Grant No.s202310130004); Inner Mongolia College Students' Innovation and Entrepreneurship Training Program Projects (Grant No.s202410130004); Inner Mongolia Natural Science Foundation (Grant No.2024LHMS06006); Inner Mongolia Natural Science Foundation (Grant No.2024MS06008); Inner Mongolia Natural Science Foundation (Grant No.2021MS06026); Scientific Research for the Public Hospitals of Inner Mongolia Academy of Medical Sciences (Grant No. 2023GLLH0211).

This paper is supported by Inner Mongolia Health Commission Project(Grant No.202201395); Baotou Municipal Health Science and Technology Project(Grant No.wsjkkj2022120); Inner Mongolia College Students' Innovation and Entrepreneurship Training Program Projects(Grant No.s202310130004); Inner Mongolia College Students' Innovation and Entrepreneurship Training Program Projects (Grant No.s202410130004); Inner Mongolia Natural Science Foundation (Grant No.2024LHMS06006); Inner Mongolia Natural Science Foundation (Grant No.2024MS06008); Inner Mongolia Natural Science Foundation (Grant No.2021MS06026); Scientific Research for the Public Hospitals of Inner Mongolia Academy of Medical Sciences (Grant No. 2023GLLH0211).

6. We note that your Data Availability Statement is currently as follows: All relevant data are within the manuscript and its Supporting Information files.

7. PLOS requires an ORCID iD for the corresponding author in Editorial Manager on papers submitted after December 6th, 2016. Please ensure that you have an ORCID iD and that it is validated in Editorial Manager. To do this, go to ‘Update my Information’ (in the upper left-hand corner of the main menu), and click on the Fetch/Validate link next to the ORCID field. This will take you to the ORCID site and allow you to create a new iD or authenticate a pre-existing iD in Editorial Manager.

8. Please amend the manuscript submission data (via Edit Submission) to include author Dr. Yuhan Qu.

9. Your ethics statement should only appear in the Methods section of your manuscript. If your ethics statement is written in any section besides the Methods, please move it to the Methods section and delete it from any other section. Please ensure that your ethics statement is included in your manuscript, as the ethics statement entered into the online submission form will not be published alongside your manuscript. 

Reviewers' comments:

Reviewer's Responses to Questions

**Comments to the Author**

1. Is the manuscript technically sound, and do the data support the conclusions?

Reviewer #1: Yes

Reviewer #2: Yes

2. Has the statistical analysis been performed appropriately and rigorously? 

Reviewer #1: Yes

Reviewer #2: Yes

3. Have the authors made all data underlying the findings in their manuscript fully available?

Reviewer #1: Yes

Reviewer #2: Yes

4. Is the manuscript presented in an intelligible fashion and written in standard English?

Reviewer #1: Yes

Reviewer #2: Yes

5. Review Comments to the Author

**Reviewer #1:**  My queries:

How does the introduction of FCGE blocks improve both computational efficiency and detection accuracy for small and blurry ground glass nodules?

Could you elaborate on how the HiLo-AIFI Module reduces computational redundancy while maintaining high performance?

Can the model be adapted to detect other types of abnormalities in medical imaging, or is it specifically tailored for ground glass nodules?

**Reviewer #2:**  It is very interesting for the manuscript entitled by "improvement of RT-DETR model for ground glass pulmonary nodule detection". However, there are still some aspects needed to be clarified for major revision. (1) The introduction should be enhanced for presenting the potential values for our proposed DL model. (2) The innovation of our proposed model should be detailedly explained in the section of Method. (3) For performance evaluation, more types of DL models should be involved for comparsion. (4) The language should be improved for better understanding.

6. PLOS authors have the option to publish the peer review history of their article (what does this mean? ). If published, this will include your full peer review and any attached files.

**Do you want your identity to be public for this peer review?** For information about this choice, including consent withdrawal, please see our Privacy Policy .

Reviewer #1: No

Reviewer #2: No

---

## [Author Response · Author response to Decision Letter 1]

2 Dec 2024

Response to Reviewer 1

1. How does the introduction of FCGE blocks improve both computational efficiency and detection accuracy for small and blurry ground glass nodules?

Response:(1)FCGE module is a new module built upon deep fusion of the FasterNet module, ELA Attention module, and ConvGLU module, which has the advantages of each module.(2) PConv in the FasterNet module applies convolutional operations to only a portion of the channels of the input features, while the other channels remain unchanged, which effectively reduces the amount of computation and memory access. (3) The ELA attention module uses strip pooling in conjunction with 1D convolution and group normalization to enhance local feature representation and improve the model's detection accuracy for small-sized targets. Specifically, 1D convolution captures localized sequence signals and pinpoint target location; group normalization improves computational efficiency; and strip pooling captures long-distance spatial dependencies and captures a wider range of feature information. (4) The ConvGLU module is a hybrid mechanism that combines Gated Linear Unit (GLU) and Depthwise Convolution (DWConv). The gating mechanism of the ConvGLU module can flexibly adjust the channel response according to the blurriness of the target edges, ensuring the model focusing on the target's key features even when the target's contour is not clear. With DWConv, ConvGLU can capture detailed features in localized areas, including edge information, which is particularly important for identifying targets with blurry edges.(5) The ablation experiments in this paper show that the detection accuracy and computational efficiency of the model are improved by adding the FCGE module.

2. Could you elaborate on how the HiLo-AIFI Module reduces computational redundancy while maintaining high performance?

Response: (1)The HiLo-AIFI module reduces computational redundancy mainly through the HiLo module, which divides the MSA layer into two paths: high-frequency attention and low-frequency attention. HiLo splits the same number of heads into two groups proportionally by assigning heads to Hi-Fi and heads to Lo-Fi, where denotes the total number of heads in the self-attention layer and the split ratio. This approach allows the model focusing on features that are more critical to the task, rather than distributing computational resources evenly across all features, thus improving computational efficiency. The HiLo module assigns features to different heads, each of which focuses on the input features, which reduces redundant computation between heads. (2)The HiLo-AIFI module remains performant with the HiLo module. Hi-Fi in the HiLo Attention module focuses on the high-frequency regions of the image, containing large amount of detailed information. It enhances the expression of local features through the mechanism of local window self-attention, which can capture the details of the internal texture of the lung nodules and the homogeneity of the internal densities of the lung nodules. This helps recognizing the differences in internal density. Hi-Fi and Lo-Fi are used to extract and analyze the features of different areas, thus maintaining the high performance of the model.

3. Can the model be adapted to detect other types of abnormalities in medical imaging, or is it specifically tailored for ground glass nodules?

Response: The model is currently only applicable to the detection of ground-glass lung nodules and is still under investigation for the detection of other abnormalities in medical imaging.

Response to Reviewers 2

Following changes has been made to the paper in response to the feedback of reviewer 2. (1) We have added one paragraph in the introduction section that shows the potential value of deep learning models in the field of lung nodule detection. (2) Summary of innovation points of each module (FCGE, Hi-Lo-AIFI, DGAK) have been added. (3) Four additional deep learning models have been added in the performance evaluation section to compare with the improved model.

---

## [Decision Letter · Decision Letter 1]

22 Dec 2024

Improvement of RT-DETR model for ground glass pulmonary nodule detection

PONE-D-24-35347R1

Dear Dr. Wang,

We’re pleased to inform you that your manuscript has been judged scientifically suitable for publication and will be formally accepted for publication once it meets all outstanding technical requirements.

Kind regards,

Kannadhasan Suriyan

Academic Editor

PLOS ONE

Additional Editor Comments (optional):

Accept with its current form

Reviewers' comments:

Reviewer's Responses to Questions

**Comments to the Author**

1. If the authors have adequately addressed your comments raised in a previous round of review and you feel that this manuscript is now acceptable for publication, you may indicate that here to bypass the “Comments to the Author” section, enter your conflict of interest statement in the “Confidential to Editor” section, and submit your "Accept" recommendation.

Reviewer #2: All comments have been addressed

Reviewer #3: All comments have been addressed

2. Is the manuscript technically sound, and do the data support the conclusions?

Reviewer #2: Yes

Reviewer #3: Yes

3. Has the statistical analysis been performed appropriately and rigorously? 

Reviewer #2: Yes

Reviewer #3: Yes

4. Have the authors made all data underlying the findings in their manuscript fully available?

Reviewer #2: Yes

Reviewer #3: Yes

5. Is the manuscript presented in an intelligible fashion and written in standard English?

Reviewer #2: Yes

Reviewer #3: Yes

6. Review Comments to the Author

Reviewer #2: Yes, this manuscirpt has been greatly improved after the initial revision. However, there are still some aspects needed to be revised.

1. The abberivations should be checked.

2. For tables, some mistakes still exists. For example, in table 1, a word "与“ is wronlgy used, etc.

3. The expression and format should be improved by a native english expert.

Reviewer #3: The authors have effectively addressed several key challenges in the detection of ground glass lung nodules, particularly regarding the accurate identification of small, irregular, and blurred edge nodules. The proposed improvements to the RT-DETR model are well-supported by novel modules, such as FCGE, HiLo-AIFI, DGAK, and the enhanced CCFF module, which tackle the issues of detection accuracy and computational efficiency.

7. PLOS authors have the option to publish the peer review history of their article (what does this mean? ). If published, this will include your full peer review and any attached files.

**Do you want your identity to be public for this peer review?** For information about this choice, including consent withdrawal, please see our Privacy Policy .

Reviewer #2: No

Reviewer #3: No
